# A Joint Subspace View to Convolutional Neural Networks

## Abstract

Motivated by the intuition that important image regions remain important across different layers and scales in a CNN, we propose in this paper a joint subspace view to convolutional filters across network layers. When we construct for each layer a filter subspace by decomposing convolutional filters over a small set of layer-specific filter atoms, we observe a low-rank structure within subspace coefficients across layers. The above observation matches widely-known cross-layer filter correlation and redundancy. Thus, we propose to jointly model filter subspace across different layers by enforcing cross-layer shared subspace coefficients. In other words, a CNN is now reduced to layers of filter atoms, typically a few hundred of parameters per layer, with a common block of subspace coefficients shared across layers. We further show that such subspace coefficient sharing can be easily extended to other network sub-structures, from sharing across the entire network to sharing within filter groups in a layer. While significantly reducing the parameter redundancy of a wide range of network architectures, the proposed joint subspace view also preserves the expressiveness of CNNs, and brings many additional advantages, such as easy model adaptation and better interpretation. We support our findings with extensive empirical evidence.

## 1 Introduction

Modern developments of deep network components (Ioffe & Szegedy, 2015; Xu et al., 2015), structures (He et al., 2016; Huang et al., 2017), and training techniques (Kingma & Ba, 2014; Zoph & Le, 2017) have enabled deep convolutional neural networks to consistently grow deeper and wider, with hundreds of millions of parameters. The trend of growing network scales leads to both severe challenges and urgent needs to fast adaptations (Finn et al., 2017; Lopez-Paz & Ranzato, 2017; Shin et al., 2017), parameter efficiency (Cheng et al., 2017; Han et al., 2015a; Luo et al., 2017; Savarese & Maire, 2019; Yang et al., 2019; Wang et al., 2018b), and interpretability (Selvaraju et al., 2017; Zhou et al., 2016). While recent works on efficient convolution operations (Chollet, 2017; Howard et al., 2017; Ma et al., 2018; Zhang et al., 2018) alleviate the long recognized over-parametrization problem of deep CNNs, and practical methods on modeling the shared components across CNN filters have been proposed (Ha et al., 2016; Savarese & Maire, 2019), there is still a lack of a principled view to exploiting the common structure of the filters within a CNN.

While preliminary studies have revealed the redundancies in deep network parameters (Michel et al., 2019; Raghu et al., 2017), and the fact that deep features across layers are highly correlated under certain linear transforms (Kornblith et al., 2019; Morcos et al., 2018; Raghu et al., 2017), exploiting this potential redundancy remains challenging. We suspect that the spatial patterns of convolutional filters, which correspond to different semantics levels, are the main reason why we cannot observe obvious cross-layer filter low-rankness. To validate our hypothesis, we decouple the spatial patterns and channel mixing of a convolutional filter by decomposing it over layer-specific 2D filter atoms, linearly combined using atom coefficients. Then with proper alignments to the layer outputs, we finally observe a low-rank structure among those atom coefficients across different layers. This highly non-trivial observation, which to our knowledge has never been reported before, hints the feasibility of enforcing a common block of atom coefficients across layers and suggests a joint subspace view to CNNs.

Applying to a wide range of network architectures, this joint subspace view leads subsequently to a novel CNN architecture in which the majority of parameters are shared across layers as a common

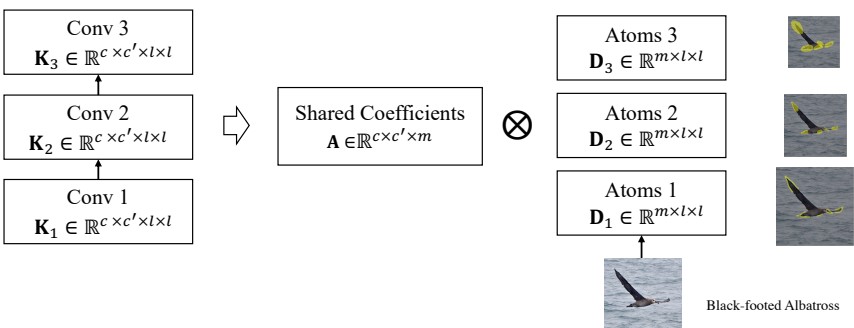

Figure 1: Atom-coefficient decomposition disentangles spatial convolution and channel mixing. So that each feature map after atom convolution now preserves spatial correspondence to the original image; and channels that describe important region features, e.g., wings, beak, and tail of a bird, will highly likely be consistently assigned with weights across layers, in order to carry on those critical information. Thus, atom coefficients, i.e., weights to linearly combine channels, now become shareable across layers to permit a joint subspace view. Such joint subspace view leads naturally to a novel CNN architecture in which the majority of parameters are shared across layers as a common block of atom coefficients, with only a few hundred parameters remaining specific to each layer as filter atoms.

block of atom coefficients, with only a few hundred parameters remaining specific to each layer as filter atoms as shown in Figure 1. This matches the intuition visualized in Figure 1 that important image regions, e.g., the wings, beak, and tail of a bird, remain important across different layers and scales to output an informative deep representation. Atom-coefficient decomposition disentangles spatial convolution and channel mixing. Thus, channels that describe important region features will highly likely be consistently assigned with large weights across layers, in order to carry on those critical information. Thus, atom coefficients, i.e., weights to linearly combine channels, now become shareable across layers to permit joint subspace CNN modeling.

Our approach accepts many easily constructed variants, e.g., using different filter atom numbers and sharing substructures, to allow highly flexible trade-offs between parameter reduction and model expressiveness. This simple and plug-and-play joint subspace view opens the door to better interpreting, training, adapting, and compacting deep models. We support our findings with extensive empirical evidence in this paper. By using variants of ACDC as plug-and-play replacements to the standard convolutions, we observe comparable and even better performance on challenging image classification datasets with orders of magnitude smaller models. We further demonstrate with few shot learning experiments that ACDC improves the adaptation of deep networks on novel tasks with limited supervisions. We end our paper with a model interpretability discussion.

Our main contributions are summarized as follows:

- We introduce a joint subspace view to CNN, which is compatible to a wide range of network architectures, to provide a structural regularization over CNN filters for better model interpretability, training, adaptation and compaction.

- Highlighting the remarkable flexibility and compatibility, we introduce various model variants constructed easily by sharing coefficient within different network sub-structures.

- We validate the effectiveness of our approach by plug-and-playing them into modern CNN architectures for various real-world tasks.

## 2   RELATED WORK

Before presenting the connections of our method with the literature, let us comment on the complementary efforts on network compression. Contrary to recent efforts on network compression and efficient architectures, we instead explore and exploit the underlying correspondence and correlations among convolutional layers within a given CNN. We decompose the convolutional kernels among layers, and enforce shared composition coefficients within sub-structures of a CNN. This

plug-and-play reparametrization of convolutional kernels significantly reduces the network redundancy while maintaining performance, improving adaptation, providing additional interpretability, reducing computational cost, and adding compression (as a by-product) comparable to many modern heavily tailored light-weight networks. We consider directions such as neural architecture search (Howard et al., 2019), post processing methods to deep networks (Phan et al., 2020; Stock et al., 2020; Li et al., 2019a; Wu et al., 2018; Son et al., 2018), and heavily tailored efficient networks and training (Tan & Le, 2019) complementary to our efforts.

**CNN architectures.** The tremendous success of applying convolutional neural networks (CNNs) on numerous tasks has stimulated rapid developments for more effective and efficient network architectures in both hand-crafted (Chen et al., 2017; He et al., 2016; Howard et al., 2017; Iandola et al., 2016; Sandler et al., 2018) and automatically discovered (Elsken et al., 2018; Liu et al., 2018; Pham et al., 2018; Zoph & Le, 2017) manners. We consider our work orthogonal to such topology-based methods, as the plug-and-play property of the proposed ACDC allows it to be added to all the aforementioned methods as a replacement to the standard convolution. Besides efforts on studying efficient network architectures, methods for network compression and pruning (Han et al., 2015a; 2016; 2015b; He et al., 2017; Li et al., 2016; Luo et al., 2017) have been extensively studied for decreasing the model size by pruning the inconsequential connections and weights of a network. Methods (Ha et al., 2016; Savarese & Maire, 2019) align with our direction as they are also insensitive to network topologies. And as shown in the experiments, ACDC can achieve higher performance in terms of parameter reduction and classification accuracy with greater flexibility.

**Kernel decomposition in CNNs** Convolutional kernel decomposition has been studied for various objectives. (Sosnovik et al., 2019) utilizes kernel decomposition as a tool for constructing same kernel with multiple receptive fields. DCFNet (Qiu et al., 2018) is proposed as a principle way of regularizing the convolutional filter structures by decomposing convolutional filters in CNN as a truncated expansion with pre-fixed bases. Split-wise decomposition over basis elements is introduced in (Li et al., 2019a) to retrain a CNN with reduced parameters.

**Weight sharing in networks** Exploiting sharing weight within CNNs is discussed in (Son et al., 2018; Wu et al., 2018) as a post-processing step for model compression. Hypernetworks (Ha et al., 2016) adopt weight generation from a shared network (denoted as Hypernet) to exploit correlations among weights empirically.

## 3 JOINT SUBSPACE CNN MODELING

In this section, we start with a brief introduction to atom-coefficient filter decomposition, which builds the foundation of our joint subspace CNN modeling (Section 3.1). We then present the empirical support for the joint subspace view to CNN filters through a motivating experiment (Section 3.2). We show in this motivating experiment that, with atom-coefficient decomposition, a jointly low-rank structure can be clearly observed among coefficients trained independently in different layers of a CNN under certain linear transformations. This observation, together with the well recognized over-parametrization problem of CNNs, leads to the idea of subspace coefficients sharing enforced across network sub-structures, which can then be constructed as plug-and-play replacements to the standard convolutions (Section 3.3 and Section 3.4).

### 3.1 CONVOLUTIONAL FILTER DECOMPOSITION

Previous works have shown that a convolutional filter in a CNN can be decomposed as a linear combination of pre-fixed basis (Qiu et al., 2018). In our approach, we adopt a similar decomposition as shown in Figure 2, in which a convolutional filter is represented as a linear combination of trainable 2D filter atoms. After decomposition, a convolution layer with $c$-channel output $\mathbf{Y}$ and $c'$-channel input $\mathbf{X}$ becomes

$$\mathbf{Y} = \mathbf{K} * \mathbf{X}, \quad \mathbf{K} = \mathbf{A}\mathbf{D}, \tag{1}$$

where $*$ denotes the convolution operation. As illustrated in Figure 2, in (1), a convolutional filter $\mathbf{K} \in \mathbb{R}^{c \times c' \times l \times l}$, which can be seen as a stack of $c \times c'$ 2D convolutional filters with the size of $l \times l$, is reconstructed by multiplying $m$ 2D filter atoms, denoted collectively as $\mathbf{D} \in \mathbb{R}^{m \times l \times l}$, with the corresponding atom coefficients $\mathbf{A} \in \mathbb{R}^{c \times c' \times m}$. Note that square filters are assumed here for simplicity, while all filter shapes are supported. Since both convolution and tensor multiplication are commutative, a convolutional layer can now be decomposed into two:

- A *atom sub-layer* where each atom involves spatial-only convolution with the filter atoms, i.e., $\mathbf{Z} \in \mathbb{R}^{c'm \times h \times w} = \mathbf{D} * \mathbf{X}$;

- A *coefficient sub-layer* that linearly combines feature channels from the *atom sub-layer*: $\mathbf{Y} \in \mathbb{R}^{c \times h \times w} = \mathbf{A}\mathbf{Z}$. Note that $\mathbf{Z}$ here denotes *atom sub-layer* outputs, and stride 1 and same padding are assumed for the sake of discussion.

## 3.2 THE MOTIVATION BEHIND

Deep CNNs are long recognized to be over-parametrized. The very deep layers in modern CNN structures (He et al., 2016; Huang et al., 2017; Zagoruyko & Komodakis, 2016) and the high-dimensional filters with little structural regularizations lead to hundreds of millions of parameters. Such over-parametrization problem is also observed in the studies of deep representations (Raghu et al., 2017), and empirically alleviated by new network structures (Chollet, 2017; Howard et al., 2017), network compression, and parameter reduction methods (Ha et al., 2016; Savarese & Maire, 2019).

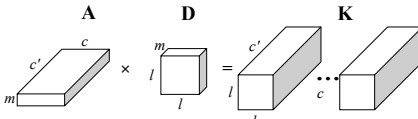

Figure 2: Atom-coefficient filter decomposition. A convolutional kernel $\mathbf{K}$ is decomposed over a set of $m$ 2D filter atoms $\mathbf{D}$ linearly combined by atom coefficients $\mathbf{A}$.

Meanwhile, recent studies on deep representations (Kornblith et al., 2019; Morcos et al., 2018; Raghu et al., 2017) have shown that there exists obvious correlations in features across layers within a CNN after proper linear transformations. The correlation of features across network layers motivates us to explore and exploit correlations across filters for structural regularizations.

We present here a motivating experiment on MNIST by applying CCA alignments as in (Raghu et al., 2017) to the *atom sub-layer* outputs and the *coefficient sub-layer* outputs of layer $i$ and layer $j$. Note that no atom coefficient sharing is yet imposed here, and the network reports the same testing accuracy before and after filter decomposition. Formally, $c$, $m$, $d$, and $hw$ denote the number of channels, number of filter atoms, test set size, and the 2D feature dimensions, respectively. The *atom sub-layer* outputs of the $i$-th and $j$-th layer, $\mathbf{Z}_i$ and $\mathbf{Z}_j \in \mathbb{R}^{cm \times dhw}$, are firstly aligned by linear transformations $\mathbf{P}_i$ and $\mathbf{P}_j \in \mathbb{R}^{cm \times cm}$ that maximize the correlation $\rho_z = \max_{\mathbf{P}_i, \mathbf{P}_j} corr(\mathbf{P}_i \mathbf{Z}_i, \mathbf{P}_j \mathbf{Z}_j)$. And similarly, the *coefficient sub-layer* outputs of both layers, $\mathbf{Y}_i$ and $\mathbf{Y}_j \in \mathbb{R}^{c \times dhw}$, are aligned by $\mathbf{Q}_i$ and $\mathbf{Q}_j \in \mathbb{R}^{c \times c}$ that maximize the correlation $\rho = \max_{\mathbf{Q}_i, \mathbf{Q}_j} corr(\mathbf{Q}_i \mathbf{Y}_i, \mathbf{Q}_j \mathbf{Y}_j)$. Omitting the layer indexes, the feed forwards of both layers can be rewritten as

$$\mathbf{Y} = \mathbf{Q}\mathbf{A}\mathbf{P}^{-1}\mathbf{P}(\mathbf{D} * \mathbf{X}). \tag{2}$$

By merging the transform into the coefficients $\mathbf{A}$ by $\widetilde{\mathbf{A}} = \mathbf{Q}\mathbf{A}\mathbf{P}^{-1}$, we obtain 'aligned coefficients' $\widetilde{\mathbf{A}}_i$ and $\widetilde{\mathbf{A}}_j$, that reside in a low rank structure reflected by the very similar effective ranks of $\widetilde{\mathbf{A}}_i$ and $[\widetilde{\mathbf{A}}_i, \widetilde{\mathbf{A}}_j]$. For example, in our MNIST experiment using a 4-layer 32-channel CNN, out of the 6 possible $(i, j)$ pairs, the average effective rank of $\widetilde{\mathbf{A}}_i$ and $[\widetilde{\mathbf{A}}_i, \widetilde{\mathbf{A}}_j]$ are 31.98 and 38.56, respectively. Our observations agree with and further support recent studies on cross-layer feature correlations (Kornblith et al., 2019; Morcos et al., 2018; Raghu et al., 2017). This observation hints that, instead of treating filters in each layers independently, it is feasible to exploit such intrinsic low-rank structure of coefficients across layers, and impose a joint subspace view to filters in a CNNs. And practically, this observation supports the feasibility of directly reducing the parameter redundancy of a CNN by enforcing shared atom coefficients across layers.

## 3.3 COEFFICIENTS SHARING ACROSS LAYERS

Based on the observations above and (1), a joint subspace CNN modeling is constructed by enforcing common atom coefficients $\mathbf{A}$ across layers in a CNN, as illustrated in Figure 1. Formally, given a $N$-layers CNN, the $n$-th convolutional filter is constructed by

$$\mathbf{K}_n = \mathbf{A}\mathbf{D}_n, \forall n = 1, \dots, N. \tag{3}$$

Assuming for now all layers have identical channel number with $c' = c$, the amount of parameters is reduced from $c^2 c^2 N$ to $c^2 m + N k l^2$.

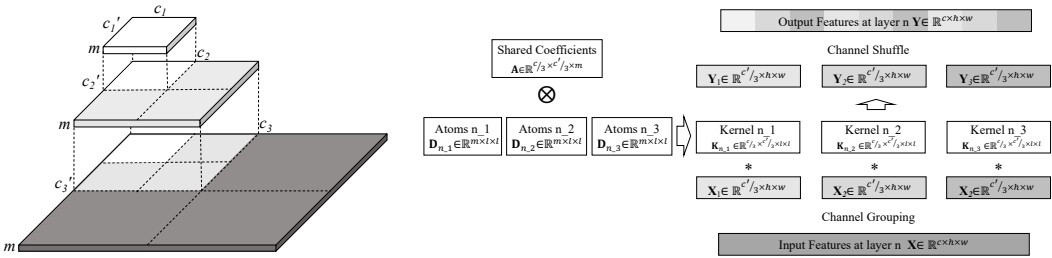

Figure 3: Illustration on how coefficients are shared across three layers with increasing numbers of channels. The shared coefficients are initialized with the largest dimensions required.

Figure 4: atom coefficient sharing with three groups at layer $n$. The input feature is first equally divided into groups (denoted as boxes with different grey scales), each of which is convolved with one group of filters reconstructed by multiplying the corresponding filter filter atoms and the shared coefficients. The output of three groups are combined by channel shuffle.

In practice, convolution layers within a network can have different numbers of channels. When sharing coefficients across layers with different channels numbers, we initialize the dimensions of the shared coefficients to be the largest dimensions needed by the corresponding layers. For example, given a $N$-layer CNN with convolutional kernels $\{\mathbf{K}_n \in \mathbb{R}^{c_n \times c'_n \times l \times l}; n = 1, \ldots, N\}$, the network is constructed by initializing the shared coefficient as $\mathbf{A} \in \mathbb{R}^{c_{max} \times c'_{max} \times m}$, where $c_{max} = \max\{c_n; n = 1, \ldots, N\}$ and $c'_{max} = \max\{c'_n; n = 1, \ldots, N\}$. The filters with fewer channels are reconstructed by multiplying the filter atoms with a subset of the shared coefficients $\mathbf{K}_n = \mathbf{A}[1 : c_n, 1 : c'_n, 1 : m]\mathbf{D}_n$. A 3-layer illustration with progressively increased channels is shown in Figure 3. Such a design choice is motivated by multi-scale decomposition (Mallat, 1999), and proves to be highly effective with our extensive experimental validation.

Note that a relaxed version can be formulated as follows: Instead of enforcing common coefficients across all layers, we allow the sharing to happen among a few consecutive layers in a network. We refer a group of consecutive layers with identical number of output channels and identical feature sizes as a *block* in a deep network, and enforce coefficient sharing in each block. For example, a VGG16 (Simonyan & Zisserman, 2014) can be implemented by sharing coefficients within 5 groups, each of which consists of conv layers with 64, 128, 256, 512, 512 channels, respectively.

### 3.4 COEFFICIENTS SHARING ACROSS FILTER GROUPS

Motivated by the observation in (Raghu et al., 2017) that the representation learned at a layer is not fully determined by the number of neurons in the layer, and the existence of parameter redundancy within a single layer as studied in (Michel et al., 2019), we suggest to further extend our joint subspace view from coefficient sharing across layers to sharing across groups of filters in a convolutional layer.

Practically, we further break down the smallest sharing unit from a layer to part of a layer. A high-dimensional convolutional layer can now be separated into several groups with identical sizes, and sharing coefficients are imposed across groups. Formally, given a convolutional layer with $c'$ input channels and $c$ output channels, respectively, we divide input channels into $g$ identical-sized groups, and each group is convolved with a convolution kernel $\mathbf{K}_j \in \mathbb{R}^{\frac{c}{g} \times \frac{c'}{g} \times l \times l}, j = 1, \ldots, g$. After grouping, we decompose $\{\mathbf{K}_j; j = 1, \ldots, g\}$ into shared coefficients $\mathbf{A} \in \mathbb{R}^{\frac{c}{g} \times \frac{c'}{g} \times m}$, and $g$ independent sets of filter atoms $\{\mathbf{D}_j \in \mathbb{R}^{m \times l \times l}; j = 1, \ldots, g\}$. In this way, the number of shared coefficients is reduced by $g^2$ times, and the number of filter atoms is increased by $g$ times. Since filter atoms have orders of magnitude smaller dimension comparing to the coefficients, we achieve further parameter reduction. Since each $\mathbf{K}_j$ only convolves with a subset of the input feature, this method reduces the overall computations.

However, directly deploying this sharing mechanism breaks the network into several paralleled sub-networks with no feature passing and gradient propagation among them. To remedy this without adding any additional parametric components, we utilize *channel shuffle* (Zhang et al., 2018) that enables information to be efficiently propagated among groups in a non-parametric way. An illustration is presented in Figure 4. Since the size of the shared coefficient now does not depend on the

largest feature dimension of a network but the size of the groups, we obtain a more compact model. In practice, to handle the expanding channels numbers in typical CNNs, we can easily allow each group in the shallow layer to feed the features to multiple groups in the deep layer. Therefore the proposed coefficient sharing across filter groups permits remarkable parameter sharing with little modification to the original network structures such as the numbers of channels and layers.

Table 1: Comparisons on CIFAR-10 with parameter sizes, parameter reduction rates, and test error. Those with higher than baseline accuracy but fewer parameters are marked in bold.

| Architectures | $m$ | $s$ | VGG16 Size | Error | ResNet18 Size | Error | WRN-40-4 Size | Error |
|---|---|---|---|---|---|---|---|---|
| Baseline | - | - | 14.72M | 6.20 | 11.17M | 5.81 | 8.90M | 4.97 |
| ACDC-net | 8 | - | 2.11M (85.7%↓) | **5.67** | 2.28M (79.6%↓) | 5.98 | 0.58M (93.5%↓) | **4.85** |
| | 16 | - | 4.21M (71.4%↓) | **5.44** | 4.38M (60.8%↓) | **5.43** | 1.11M (87.5%↓) | **4.42** |
| ACDC-block | 8 | - | 4.89M (66.8%↓) | **5.47** | 2.96M (73.5%↓) | **5.50** | 0.74M (91.7%↓) | **4.46** |
| | 16 | - | 9.78M (33.6%↓) | **5.40** | 5.76M (48.4%↓) | **4.92** | 1.43M (83.9%↓) | **4.38** |
| ACDC-g-net | 8 | 32 | 0.03M (99.8%↓) | 10.24 | 0.20M (98.2%↓) | 7.30 | 0.07M (99.2%↓) | 8.20 |
| | 16 | 64 | 0.08M (99.5%↓) | 9.87 | 0.26M (97.7%↓) | 7.91 | 0.13M (98.5%↓) | 6.85 |
| ACDC-g-block | 8 | 32 | 0.06M (99.6%↓) | 9.71 | 0.22M (98.0%↓) | 7.48 | 0.09M (99.0%↓) | 8.92 |
| | 16 | 64 | 0.35M (97.6%↓) | 6.63 | 0.45M (96.0%↓) | 7.22 | 0.26M (97.1%↓) | 6.88 |
| ACDC-g-layer | 8 | 32 | 0.13M (99.1%↓) | 6.68 | 0.89M (92.0%↓) | 5.23 | 0.36M (96.0%↓) | 5.02 |
| | 16 | 64 | 0.80M (94.6%↓) | **5.67** | 0.60M (94.6%↓) | 6.20 | 1.98M (77.8%↓) | **4.23** |

## 4 EXPERIMENTS

In this section, we support our findings with extensive empirical evidence, and evaluate variants of our joint subspace CNN modeling, referred to as ACDC, as plug-and-play replacements to the standard convolution, with several different levels of atom coefficient sharing:

- *ACDC-net*: Sharing across all layers.
- *ACDC-block*: Sharing across blocks of layers.
- *ACDC-g*: Sharing across filter groups.

*ACDC-g* naturally further leads to three variants by allowing sharing within the entire network, blocks within a network, and layers within a network, and are thus named as *ACDC-g-net*, *ACDC-g-block*, and *ACDC-g-layer*, respectively. We use $m$ and $s$ to denote the number of filter atoms and grouping size, respectively.

### 4.1 IMAGE CLASSIFICATION

In this section, we perform standard image classification experiments, and validate our joint subspace CNN modeling with extensive experiments on CIFAR-10, CIFAR-100, *Tiny*ImageNet, and ImageNet.

**Evaluating variants of ACDC.** We first report on CIFAR-10 self-comparisons on variants of ACDC constructed with different numbers of filter atoms as well as grouping sizes. We present performance in terms of both parameter size and classification error in Table 1. VGG16 (Simonyan & Zisserman, 2014), ResNet18 (He et al., 2016), and Wide ResNet (WRN) (Zagoruyko & Komodakis, 2016) are adopted as the underlying network architectures in order to show the remarkable compatibility of the proposed approach. This set of experiments show our joint subspace modeling maintains or improves CNN performance with a significantly more compact model, e.g., 98% reduction with comparable performance, and 70% reduction with even higher accuracy.

**CIFAR-10, CIFAR-100 and *Tiny*ImageNet.** We further present experiment results on CIFAR-10, CIFAR-100, and *Tiny*ImageNet in Table 2. We compare exampled variants of ACDC against HyperNetworks (Ha et al., 2016) and Soft Parameter Sharing (Savarese & Maire, 2019), both of which serve as plug-and-play replacements to standard convolutions as well. Though HyperNetworks (Ha et al., 2016) achieves remarkable parameter reduction, ACDC is able to achieve higher accuracies

Table 2: Classification performances on CIFAR-10, CIFAR-100, and *Tiny*ImageNet datasets. Performance on state-of-the-art light CNN architectures are listed in the upper block. The middle block shows the performance of plug-and-play methods with parameter sharing in CNNs. We present the results of variants of ACDC in the bottom block. Performance obtained by our reproductions are marked with ∗.

| Methods | Size | CIFAR-10 | CIFAR-100 | *Tiny*ImageNet |
|---|---|---|---|---|
| SqueezeNet (Iandola et al., 2016) | 2.36M | 6.98∗ | 29.56∗ | 48.22∗ |
| ShuffleNet (Zhang et al., 2018) | 0.91M | 7.89∗ | 29.94∗ | 54.72∗ |
| ShuffleNet-V2 (Ma et al., 2018) | 1.3M | 8.96∗ | 29.68∗ | 51.13∗ |
| MobileNet-V2 (Sandler et al., 2018) | 2.36M | 5.52∗ | 30.02∗ | 48.22∗ |
| NASNet (Zoph et al., 2018) | 3.1M | 3.59 | 21.77∗ | 47.17∗ |
| DCFNet (K=6, FB) VGG16 (Qiu et al., 2018) | 9.84M | 5.98∗ | 32.75∗ | 57.70∗ |
| Learning filter bases VGG16 (Li et al., 2019b) | 3.21M | 6.23 | - | - |
| LegoNet-VGG16-w(o=4,m=0.25) (Yang et al., 2019) | 0.9M | 8.65 | 30.11∗ | 55.22∗ |
| LegoNet-VGG16-w(o=2,m=0.5) (Yang et al., 2019) | 3.7M | 6.77 | 29.45∗ | 48.42∗ |
| WRN-40-1 HyperNets (Ha et al., 2016) | 0.10M | 8.02 | - | - |
| WRN-40-2 HyperNets (Ha et al., 2016) | 2.24M | 7.23 | - | - |
| SWRN 28-10-1 (Savarese & Maire, 2019) | 12M | 4.01 | 19.73 | 43.05∗ |
| SWRN 28-10-2 (Savarese & Maire, 2019) | 17M | 3.75 | 18.37 | 41.12∗ |
| VGG16 ACDC-g-layer $m8$ $s32$ | 0.13M | 6.68 | 28.81 | 49.96 |
| VGG16 ACDC-net $m8$ | 2.11M | 5.67 | 22.13 | 43.34 |
| WRN-40-1 *ACDC-block* $m8$ | 0.043M | 7.19 | 30.23 | 51.47 |
| WRN-40-1 *ACDC-block* $m24$ | 0.114M | 7.02 | 28.14 | 49.05 |
| WRN-40-4 *ACDC-g-layer* $m16$ $s32$ | 0.67M | 4.38 | 20.04 | 45.87 |
| WRN-28-10 *ACDC-g-block* $m24$ $s160$ | 2.27M | 4.25 | 19.64 | 41.24 |
| WRN-28-10 *ACDC-net* $m12$ | 5.21M | 3.52 | 18.81 | 39.96 |
| WRN-28-10 *ACDC-block* $m24$ | 13.20M | **3.26** | **17.85** | **38.74** |

Table 3: Performance on ImageNet. Parameters, top-1 and top-5 errors are reported. Numbers obtained by our reproductions are marked with ∗. We use ResNet34 and ResNet50 as baselines.

| Methods | Size | Top-1 | Top-5 |
|---|---|---|---|
| ResNet34 | 21.28M | 27.42∗ | 9.02∗ |
| Structured Conv A | 9.82M | 27.19 | - |
| Structured Conv B | 5.60M | 30.56 | - |
| *ACDC-g-layer* $m12s64$ (ours) | 1.66M | 32.82 | 12.14 |
| *ACDC-net* $m12$ (ours) | 3.34M | 30.85 | 11.25 |
| *ACDC-stage* $m16$ (ours) | 5.77M | 27.78 | 9.72 |
| ResNet50 | 25.26M | 24.17∗ | 7.82∗ |
| ChPrune | 17.89M | 27.7 | 9.2 |
| LegoNet-w | 9.3M | - | 8.7 |
| Versatile | 11.0M | 25.5 | 8.2 |
| Structured Conv A | 13.49M | 24.35 | - |
| Structured Conv B | 8.57M | 26.59 | - |
| *ACDC-net* $m8$ (ours) | 14.29M | 25.38 | 8.08 |
| *ACDC-stage* $m8$ (ours) | 16.37M | 24.04 | 7.68 |

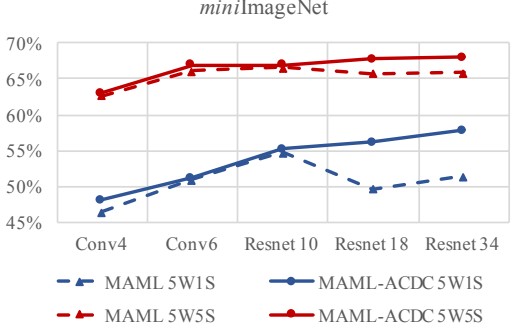

Figure 5: Few-shot image classification with deeper CNN architectures. 5W1S and 5W5S denote 5-way 1-shot and 5-way 5-shot experiments, respectively. Performance (Y-axis) is evaluated by averaging 3,000 rounds of randomly sampled testing tasks.

with even fewer parameters. The parameter reductions in Soft Parameter Sharing (Savarese & Maire, 2019) are highly restricted by the large scale elements in the filter bank. For example, SWRN 28-10-1, as the smallest variant of Soft Parameter Sharing on WRN, adopts a single template per sharing group, and can only achieve 66% of parameter reduction. By adopting *ACDC-net* and *ACDC-block* to WRN, we are able to achieve both higher parameter reductions and accuracy. Though model compaction is not our mere objective, we include here as references some state-of-the-art light CNN architectures (Iandola et al., 2016; Zhang et al., 2018; Ma et al., 2018; Sandler et al., 2018) and architecture based on neural architecture search (Zoph et al., 2018). Additional comparisons against network compression and pruning methods are in supplementary material Section C.

**ImageNet.** To fully validate our joint subspace CNN modeling, we perform further experiments on the large-scale ImageNet dataset (Deng et al., 2009). We use ResNet34 and ResNet50 as baseline models, and compare ACDC against LegoNet (Yang et al., 2019), Versatile (Wang et al., 2018b),

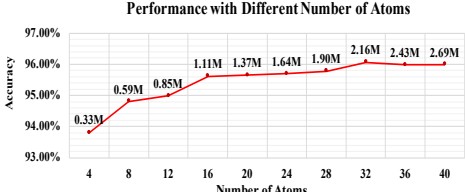
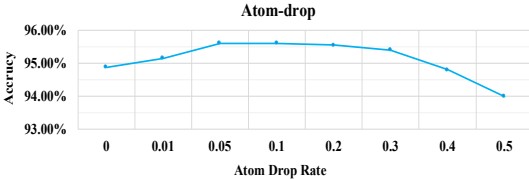

Figure 6: Accuracy with different number of filter atoms $m$. Parameter sizes are denoted as $\#M$.

Figure 7: Accuracy with different atom drop rate $p$.

network pruning method ChPrune (He et al., 2017), and recently proposed Structured Conv (Bhalgat et al., 2020). Results are presented in Table 3. For ResNet34, one variant of ACDC with grouping in layers uses only 1.66 million parameters with acceptable performance decrease. For ResNet50, since a large proportion of the parameters are in the $1 \times 1$ conv layers, ACDC achieves fair parameter reductions with maintained performance.

## 4.2 FEW-SHOT EXPERIMENTS

We further demonstrate that our joint subspace modeling improves deep model adaptation on novel tasks with limited supervisions, which is validated by few-shot classification using commonly adopted experimental settings. Specifically, we adopt *ACDC-net* on the model-agnostic meta-learning (MAML) (Finn et al., 2017) algorithm, which is a method that adapts to a novel task by tuning the entire network from a learned initialization. Although MAML is designed to be model-agnostic, we consistently observe that it struggles for further performance improvements when using deeper and wider networks. Same observations are reported in (Chen et al., 2019). We show that such limitation can be alleviated by structural regularizations with ACDC. We follow the same experimental settings as (Chen et al., 2019) and perform both 5-way 1-shot and 5-way 5-shot image classifications on *mini*ImageNet dataset. The comparisons are shown in Figure 5. Though adopting ResNet10 with MAML achieves improvements over simple network with few stacked layers, the performance drops with more residual layers as shown by the results on ResNet18. By using *ACDC-net* with deeper ResNets, performance is not only maintained but also improved when more layers are used, which thus permits larger models in MAML.

Table 4: Comparisons of FLOPs and speed with different variants of ACDC with grouping.

| Networks | Baseline | *ACDC m8 s64* | *ACDC m16 s64* | *ACDC m8 s32* | *ACDC m16 s32* |
|---|---|---|---|---|---|
| VGG16 | 125.66B | 64.15B | 64.17B | 32.29B | 32.30B |
| ResNet18 | 222.4B | 116.76B | 116.78B | 60.13B | 60.14B |
| Speed (ms/batch) | 8.9 | 5.8 | 5.8 | 4.2 | 4.3 |

## 4.3 COMPUTATIONAL EFFICIENCY

ACDC enjoys another merit of being computationally efficient when using *sharing with grouping*. Since after grouping, each group of convolutional filters only convolves with a subset of input features, *ACDC-g-block* and *ACDC-g-net* substantially reduce the number of FLOPs and accelerate the speed. We report comparisons with ResNet18 and VGG16 in Table 4. All numbers are obtained by feeding the network a typical batch with 100 $64 \times 64$ images. It is clearly shown that by using small groups, the computation can be reduced dramatically, and larger number of filter atoms only effects the speed the total FLOPs slightly. We present further speed comparisons against recent efficient network structures in Appendix Table B.

**Number of filter atoms** One additional hyperparameter introduced is the number of filter atoms $m$ per sub-structure. As shown in Figure 6, having more filter atoms in each sub-structure leads to performance improvements that saturate at $m = 32$. More filter atoms also result in larger parameter sizes, which are unfavourable.

**Regularization by atom-drop.** To improve the robustness of the filter atoms and the corresponding reconstructed kernels, we further propose a structural regularization to filter atoms named *atom-drop* inspired by widely used dropout. Specifically, when training the network, we randomly drop a filter atom with probability $p$, which is referred as *atom drop rate*, by temporarily setting values of the dropped filter atoms to $0$, and the values of all other remained filter atoms are multiplied by $\frac{1}{1-p}$ in order to maintain consistent scales of the reconstructed convolutional kernels. At test time, all filter atoms are presented with no dropping. As shown in Figure 7, atom-drop improves generalization when $p \le 0.1$. Higher values of $p$ potentially degrade the performance as the training becomes unstable. Thus we use $p = 0.1$ as the default setting. We show in Appendix Section B further comparison against typical dropout methods.

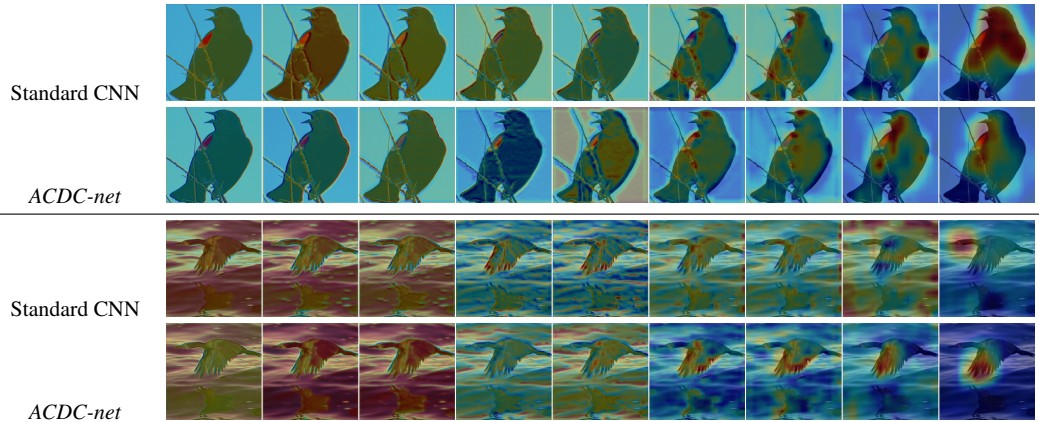

Figure 8: Illustration on extending CAM to all layers (shallow $\rightarrow$ deep) with standard CNN and *ACDC-net*. While CAM is originally introduced to explain the feature at the final convolutional layer, we show that sharing coefficients allows CAM to better explain the shallow layers. The network is progressively extracting features that attend to the discriminative regions, e.g., the wings or the head of a bird.

## 5 CONCLUSION AND INTERPRETABILITY DISCUSSION

In this paper, we introduced a joint subspace approach to CNNs. We presented observations that, due to the underlying cross-layer correlations, atom coefficients in the decomposed convolution layers across a CNN reside in a low-rank structure. We explicitly exploited such observations by enforcing atom coefficients to be shared within sub-structures of CNNs. Variants of the proposed method can be constructed with different sharing structures, number of atoms, and grouping sizes. We reported extensive experiment results that show the effectiveness of the proposed method on standard image classification and adaptations.

This joint subspace modeling has the potential for better interpretability of CNNs, due to the cross-layer shared coefficients. Class activation mapping (CAM) (Zhou et al., 2016), is a method that is originally proposed to explain the importance of image regions **only at the final convolution layer**, and is calculated by weighted averaging the features of the final convolution layer by the weight vector of a particular class. We close our paper with an illustration in Figure 8 that **extends class activation mapping (CAM) to all layers** of a ResNet-18 and a ResNet-18 with *ACDC-net*. The networks are trained from scratch on CUB-200 (Wah et al., 2011) high-resolution ($448 \times 448$) fine-grained bird classification dataset.

It is clearly shown that, in *ACDC-net*, while the class activation maps for shallow layers are inevitably noisy due to the limited receptive fields, features in deeper layers are progressively refined to the discriminative regions. However, in standard CNNs, due to no explicit correspondence among filters across layers, CAM can only explain the feature of the last Conv layer. This shows the great potential for better interpretability with ACDC, and we will keep this as a direction of future effort.

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

APPENDIX

## A  IMPLEMENTATION DETAILS

All experiments are conducted on a server with 8 Nvidia RTX 3090 graphic cards, and each has 24GB memory. Experiments with few-shot image classification and CIFAR trained and tested on a single card. Experiments with ImageNet are trained with 8 cards in parallel. The machine is also equipped with 512GB memory and two AMD EPYC 7502 CPUs.

All experiments are performed using PyTorch (Paszke et al., 2019).

**Initialization.**  We train every ACDC network from scratch. We use orthogonal initialization for all filter atoms, and Kaiming normal initialization for all coefficients.

**Training details.**  All networks for CIFAR10 and CIFAR100 are trained for 350 epochs. The initial learning rate is set to be $0.1$. The learning rate decays by a factor of $10$ at the 150-th and the 250-th epoch. All ACDC networks are trained with a weight decay of $10^{-4}$. The reported numbers are averaged over 5 runs. Networks for ImageNet are trained for 90 epochs. The initial learning rate is $0.1$ and decays by a factor of $10$ every 30 epochs. The weight decay for ImageNet experiments are set to be $2.5 \times 10^{-5}$. We consistently observe that lower values for weight decay yield better results on ACDC networks. We believe the reason is that the structural regularization of ACDC already reduces the risk of overfitting.

## B  COMPARISONS OF ATOM-DROP

The proposed atom-drop shares the similar motivation with dropout, which is introduced to improve the robustness of networks weights by randomly closing activations in the training of the the deep network. Atom-drop is introduced in ACDC as a method to improve the robustness of the learned atoms at each layer. Most importantly, since atom-drop is performed on atoms, which are network parameters, and dropout is performed on feature activations, they can be applied simultaneously. We present in Table A the performance of ACDC-net m16 with WRN-40-4 and different dropping regularizations. Dropout-1d and dropout-2d correspond to randomly dropping pixels and feature channels, respectively. Using drop-atom alone delivers better performance comparing to both dropout-1d and dropout-2d, and drop-atom is compatible with dropout-1d and dropout-2d for better performance.

Table A: Performance comparisons with different dropping regularizations.

|       | dropout-1d | dropout-2d | atom-drop | atom-drop + dropout-1d | atom-drop + dropout-2d |
|-------|------------|------------|-----------|------------------------|------------------------|
| Error | 4.92       | 4.90       | 4.85      | 4.83                   | 4.82                   |

## C  COMPARISONS AGAINST NETWORK COMPRESSION AND PRUNING METHODS

We present here further comparisons against network compression and pruning methods. Performance is measured with error rates and proportion of remained parameters (both lower the better). Note that different from post-processing using compression and pruning, ACDC is primarily proposed as a structural regularization and a plug-and-play replacement to standard convolutional layers, so the networks with ACDC remain trained end-to-end. We include SNIP (Lee et al., 2018), TR (Wang et al., 2018a), and recently proposed T-Basis (Obukhov et al., 2020) and GraSP (Wang et al., 2020) into the comparisons. Results shown in Figure A clearly demonstrate that ACDC can achieve even smaller parameter size with negligent scarification to accuracy.

## D  COMPARISONS ON SPEED

We present the comparisons on speed against some recent efficient network structures in Table B. All numbers are obtained by an average of 10,000 runs. Each run contains a batch of 100 samples

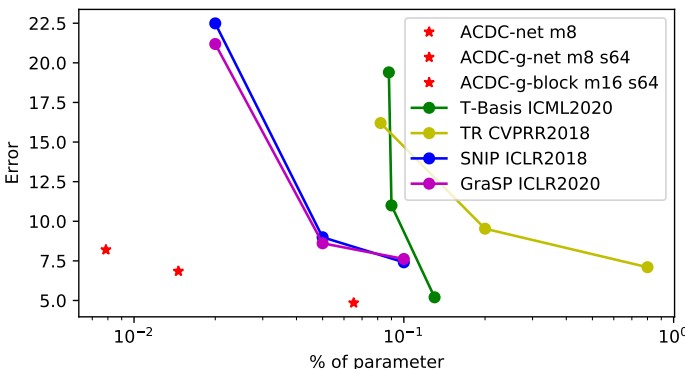

Figure A: Comparisons against network compression and pruning methods on CIFAR-10 dataset. Performance is measured with error rates and proportion of remained parameters (both lower the better).As a plug-and-play method, ACDC can outperform network compression and pruning methods in terms of both parameter reduction rate and prediction accuracy.

with a resolution of $224 \times 224$, which saturate the GPU cores. While fast speed is not the primary goal of the paper, our joint subspace modeling allows comparably fast speed with the state-of-the-art efficient network structures.

| Networks | Speed (ms) |
|---|---|
| ResNet18 (He et al., 2016) | 25.8 |
| MobileNet-v3 (Howard et al., 2019) | 23.4 |
| ShuffleNet (Zhang et al., 2018) | 13.3 |
| SqueezeNet (Iandola et al., 2016) | 25.9 |
| ResNet-18 ACDC m16 s64 | 18.6 |
| ResNet-18 ACDC m16 s32 | 16.4 |

Table B: Comparisons on speed against some recent efficient network structures.

| Training stages | 1. *ACDC-net m=8* | 2. *ACDC-block m=8* | 3. *ACDC-block m=12* |
|---|---|---|---|
| Parameters | 3.57 | 4.59 | 6.74 |
| Error | 41.76 | 40.28 | 39.06 |

Table C: Experiments with expanding network capacity.

# E   ARCHITECTURE ADAPTIVITY

ACDC offers a variety of variants with varying degrees of sharing and performance. The selection of the variants can be determined based on the factors like the size and difficulty of the datasets at hand. Furthermore, the shared atom-coefficients decomposition for all ACDC variants allows for easy transfer and, as a result, an adaptive model adaptation scheme. For example, one can start training the network with coefficients sharing across all layers (*ACDC-net*) with a relatively small number of atoms. And the network capacity can be gradually increased by relating the coefficient sharing to blocks only and adding more atoms. Meanwhile, the majority of the parameters can be inherited from the earlier trained model to guarantee training efficiency. To show this, we use WideResNet (WRN-28-10) as the backbone and train an expanding model on the *Tiny*ImageNet dataset. As shown in Table C, in stage 1, we start with training *ACDC-net m=8*. When further performance improvement is desired, we can relax the sharing by using the parameters obtained with *ACDC-net m=8* to initialize *ACDC-stage m=8* and continue training the network at stage 2. Similarly, we can further expand the network to *ACDC-stage m=12* with most of the parameter inherited from *ACDC-stage m=8*.

