# OpenReview forum: "A Joint Subspace View to Convolutional Neural Networks"
_ICLR.cc/2022/Conference — ICLR 2022 Submitted_

### Official Review · Reviewer_vUb9 · 2021-10-20

**Correctness:** 4
**Technical Novelty And Significance:** 2
**Empirical Novelty And Significance:** 2
**Recommendation:** 6
**Confidence:** 5

**Main Review:**

Depending on the low-rank structure observation, this paper proposes to jointly model filter subspace across different layers by enforcing cross-layer shared subspace coefficients. The key idea is that all convolutional filters are decomposed into a shared coefficient submatrix and special filter atoms.

+ves:
+ The idea of the shared subspace coefficient for reducing the parameter redundancy seems interesting.
+ This paper is well-written and is easy to follow.

Concerns:

- The novelty of this paper is weak. In fact, the idea that the convolutional filter is considered a multiplication of two small (low-dimensional) submatrixes has been proposed in [ref-1]. Of course, this paper proposed a shared coefficient submatrix. However, this novelty is not enough for the standard of ICLR.
- The experimental results are not surprising. For the small datasets (e.g., CIFAR-10, CIFAR-100, and TinyImageNet), they only need a small deep model essentially although the proposed method is to reduce a large number of the parameters. For the large datasets (e.g., ImageNet), the proposed method is to reduce a small number of the parameters. Moreover, many related works (e.g., ShuffleNet-V2,MobleNet-V2) are missing to compared with the proposed method on ImageNet.
-I am sincerely hoping that the authors deeply think more important problems in the topic of speeding up deep networks.

[ref-1] Speeding up Convolutional Neural Networks with Low Rank Expansions, BMVC 2014.


**Summary Of The Paper:**

To reduce the parameters of the deep networks, this paper proposes a joint subspace view to convolutional filters across network layers. Experimental results verify the effectiveness of the proposed method.

**Summary Of The Review:**

I recommend the borderline rejection due to the limited novelty and unsurprising experimental results.

---

> ### Author Response · Authors · 2021-11-16
> **Thank you for your insightful feedback!**
>
> Thanks for your constructive feedback! We have discussed and highlighted the novelty aspects of this paper in the response to all reviewers above. And we hope our responses below could further address your concerns.
>
> **1. Comparison against [a]**
>
> We would like to highlight that our contribution introduced in this paper can be orthogonal to that of [a].
> [a] explores low-rank factorization to the convolutional filter at each layer, whereas the main contribution of our paper is to discover the cross-layer filter redundancy, and exploit such intrinsic redundancy by introducing across sub-layer coefficient sharing.
> Although the basic operations in both methods are multiplications of sub-matrices, as we explained in our response to all reviewers, the novelty of our method comes from the fact that it introduces **a joint subspace view to CNNs** that takes advantage of the inherent cross-layer channel correspondence, allowing for better performance with fewer layer-specific parameters and thus more parameter savings. All the quantitative results further support the novelty of our paper.
>
> In Table 3, we compare plug-and-play replacements for convolutions only using two backbone architectures (ResNet34 and ResNet50). As a result, highly customized architectures such as ShuffleNets and MobileNets are not discussed. In the upper block of Table 2, we present the performance of lightweight networks mainly to show that by applying the proposed ACDC to existing common network structures, we can achieve the same level of parameter reduction and better performance as those highly customized architectures.
>
> **2. Experimental results**
>
> While we agree that those small datasets can be expected to solve with relatively small networks, we highlight that, as discussed in Q1 to all reviewers, ACDC provides a much wider range of performance-efficiency trade-offs compared to other methods. And it can reduce the parameters of a VGG network to **0.4%** of the original model, while still delivering over **90%** accuracy on CIFAR-10.
>
> **3. Speeding up CNNs**
>
> Thanks for your insightful suggestion. We agree that speeding up CNNs is a critical task and we have presented some preliminary efforts on speeding up CNNs in Table 4. We will further explore this in our future work.
>
> [a] Speeding up Convolutional Neural Networks with Low Rank Expansions, BMVC 2014.

---

> ### Comment · Area_Chair_VSUA · 2021-11-26
> **Please respond to the author rebuttal**
>
> Dear Reviewer vUb9,
>
> The authors have posted their rebuttal. I wonder whether the rebuttal addressed your concerns? Please respond to the authors. Thanks!
>
> AC

---

### Official Review · Reviewer_r8DN · 2021-11-01

**Correctness:** 3
**Technical Novelty And Significance:** 3
**Empirical Novelty And Significance:** 3
**Recommendation:** 6
**Confidence:** 3

**Main Review:**

Pros
1.	Motivation is well defined. By suggesting motivating experiment, the method of the paper could be verified theoretically as well as experimentally.
2.	The method is simple but very strong. The method is just decomposing the shared atom coefficient of the layers in CNN, then they convolute images only with layer-specific coefficients followed after matrix-vector multiplication with respect to shared coefficients. Furthermore, by using group convolution, they accelerate the speed of the algorithm as in Tab.4. Even with fast computation, it shows good performance at network reduction, with at least maintaining or improving the error ratio.
3.	The method can be compatible with various case. Since there are additional methods to be combined with the proposed such as block (section 2.3.) and group (section 2.4.) approaches, it can have a lot of variants.

Cons
1.	There is a lack of reasons for using maximum value of C_in and C_out in the section 2.3. If you already have ablation study on that, it would be better to have in text, somewhere.

Minors
1.	The caption of Figure 1 is same as the third paragraph of the Introduction (replica). If the authors provide additional description of the figure, it will be clearer.
2.	The authors need to explain about the bottom block in table 2. The description of third paragraph of section 3.1 has no referred table or figure. (The former is likely results about your methods and the latter seems to be an analysis about table 2.)  Also, the author needs to clarify the meaning of y-axis in figure 5.
3.	There is a typo in Fig. 4 : D_{n_1}\in\mathbb{R}^{k\times l\times l} -> D_{n_1}\in\mathbb{R}^{m\times l\times l}
4.	The organization of the manuscript is uncommon; a section of related work is located after result section.


**Summary Of The Paper:**

This paper introduces a joint subspace view between the layers of convolutional neural network. Since the important features across the layers in deep CNN is maintained, the filters can be decomposed into shared coefficients and layer-specific coefficients. Motivated by multi-scale decomposition, they compose the shared coefficient in order to exploit correlations between features maximally. Also, by using group convolution, they reduce the amount of parameters as well as maintain the performance. This method is easily compatible to modern CNN architectures.

**Summary Of The Review:**

Overall, the paper is hard to follow the logic because of unusual organization. However, the novelty of the proposed method is fully convinced with supported experiments and their explanations.

---

> ### Author Response · Authors · 2021-11-16
> **Thank you for your thorough review!**
>
> We appreciate your constructive feedback and the support on the novelty of our work. We have revised the paper accordingly and hope our responses below will address your concerns fully.
>
> **1. The caption of Figure 1**
>
> We have revised the caption of Figure 1 in the revision.
>
> **2. Further explanations and reference to tables**
>
> Thanks for your thorough review. We have updated the captions of Table 2, Figure 5, and the 3rd paragraph of Section 4 as you suggested.
>
> **3. Typos**
>
> We sincerely appreciate your careful reviews. The typo in Figure 4 has been fixed in the revision.
>
> **4. Organization of the discussion**
>
> We have moved the Related Work section to the second section. We appreciate your suggestions and hope this reorganization can help improve the logic flow of our paper.
>
> **5. The size of the shared coefficients**
>
> We introduce how to perform coefficient sharing across the entire network in Section 2.3 (current Section 3.3), therefore the size of the shared coefficients should be at least not smaller than the maximum size of the coefficiets of all layers, which correspondes to $\mathbf{A} \in \mathbb{R}^{c_{max} \times c^\prime_{max} \times m}$, where $c_{\text{max}} = \max\{c_n; n=1,\dots, N \}$ and $c_{\text{max}}^\prime = \max\{c_n^\prime; n=1,\dots, N \}$. This is required by the design of our method rather than an empirical selection.

---

> > ### Comment · Reviewer_r8DN · 2021-11-29
> > **Thanks for the responses**
> >
> > Thanks for the authors' effort on clarification of my queries about dimensions. Still, I think the novelty of this paper is still in a range of incremental contribution, so I stick to my previous recommendation.

---

> ### Comment · Area_Chair_VSUA · 2021-11-26
> **Please respond to the author rebuttal**
>
> Dear Reviewer r8DN,
>
> The authors have posted their rebuttal. I wonder whether the rebuttal addressed your concerns? Please respond to the authors. Thanks!
>
> AC

---

### Official Review · Reviewer_FvfC · 2021-11-01

**Correctness:** 3
**Technical Novelty And Significance:** 2
**Empirical Novelty And Significance:** 3
**Recommendation:** 6
**Confidence:** 4

**Main Review:**

Strengths:
(1)	The idea of coefficient sharing across different settings (e.g., the whole networks, the convolution stages, convolution layers, and groups of filters) seem interesting.
(2)	The experimental results show the proposed ACDC is superior or competitive to state-of-the-art methods.
(3)	The paper is clear written.

Weaknesses:
(1)	My first concern is the settings of coefficient sharing. For CNN models, different convolution stages capture various kinds of features. For example, the preceding layers or blocks extract the fundamental cues (e.g., color or boundaries), while the latter layers or blocks learn the high-level semantic features. Although there exist some correlations in features across layers, correlations among different-level settings may be clearly different. Therefore, the best settings (e.g., the whole networks, the convolution stages, convolution layers, and groups of filters) of coefficient sharing may vary in various backbones and tasks. So, could the authors give the guideline or develop an adaptive method?
(2)	As shown in Table 1, coefficient sharing across groups of filters have great effect on the final performance. Could the authors give more detailed discussion? Furthermore, the original backbone models with only groups of filters could be compared as baselines for verifying the effectiveness of ACDC-g. How about the performance of ACDC-g on ImageNet?
(3)	The authors claim that the proposed method can provide better interpretation for CNN. However, it seems not very clear for me. Could the authors give more detailed discussion?

Other Comments:

1.	Compared to 2D convolutions, 3D convolutions need more optimization due to high computation complexity. Therefore, do the proposed method adopt to 3D convolutions (spatial and temporal information are coupled)?  Could the authors provide some tiny examples about 3D convolutions?
2.	Some typos, e.g., 13.49-> 13.49M and 8.57 -> 8.57M in Table 3.


**Summary Of The Paper:**

This work first shows empirical observations that there exist obvious correlations in features across layers within a CNN after proper linear transformations. Accordingly, the authors propose coefficient sharing strategies across layers or across groups of filters, which aim to decompose convolution filters for reducing model sizes. The experiments are conducted on four image benchmarks using several CNN backbones, and the results show the effectiveness of the proposed ACDC on image classification and few-shot image classification tasks.

**Summary Of The Review:**

The idea of coefficient sharing across different settings seem interesting, and a ok experimental results. However, there exist some weaknesses in this work (see main comments above). Therefore, my current rating is borderline accept.

---

> ### Author Response · Authors · 2021-11-16
> **Thank you for your supportive feedback!**
>
> We are grateful for your thorough review and valuable comments. We are glad that you appreciate the idea of this paper, and hope our responses will address your concerns fully.
>
> **1. How to determine the level of sharing**
>
> The main goal of this paper is to reveal CNNs' inherent cross-layer low-rankness and exploit it by exploring coefficient sharing and introducing atom-coefficient decomposition convolutional layers as **plug-and-play replacements** for standard convolutional layers, which are shown to offer a wide range of parameter reduction and performance trade-offs.
> The choice of different variants of ACDC can be driven by the performance of particular deployment scenarios. For example, one could begin by training ACDC-net, which reduces the number of parameters significantly, and then gradually relax the sharing by using the learned cross-network coefficients to initialize ACDC-block and fine-tuning the parameters for even better performance.
> Meanwhile, without modifying the network structures, additional atoms can also be initialized and included in the training.
> To demonstrate this, we have included some preliminary results of this adaptive idea in the revision's **Appendix Section E**.
> And we will keep this as a direction of future efforts.
>
> **2. ACDC-g**
>
> Compared to ACDC-net and ACDC-block, ACDC-g admits further sharing and parameter reduction by sacrificing the fully connections across channels between layers, therefore cannot achieve the same level of network performance.
> In Table 3 of the original manuscript, we have included the ImageNet result of applying ACDC-g to ResNet-34 (*ACDC-layer m12 s64 (ours)*), demonstrating that a super small scale network (1.66M) can still perform well on ImageNet, which is never observed before to the best of our knowledge. To avoid confusion, we have modified (marked in blue) *ACDC-layer m12 s64 (ours)* to *ACDC-g-layer m12 s64 (ours)* in Table 3 of revision.
>
> **3. Interpretability**
>
> As we have shown in Section 5, typical methods of visualizing the activation maps grounding the classification can only be applied to the final convolution layer, which is usually the only layer that is connected to the final global average pooling and linear classification layer.
> Even though there might exist channels at different layers that correspond to the same visual clues, such correspondence can only be revealed after certain permutation and linear transformation [b], therefore applying techniques like [a] directly to intermediate feature layers can not yield any meaningful interpretations.
> The proposed ACDC-net, on the other hand, uses shared coefficients across the entire network to ensure that the channels are aligned across layers. As shown in Figure 8, the natural alignments achieved by our methods allow [a] to be applied directly to all layers of the deep CNN, resulting in progressive visualization of the visual clues that ground the classification from shallow to deep layers.
>
> **4. ACDC + 3D Convolution**
>
> Because the decomposition and sharing scheme proposed in this paper is agnostic to the number of dimensions, extending ACDC to 3D convolutions is simple, and benefits even more due to the potentially greater redundancy in 3D convolutions. To demonstrate this, we use ACDC to train an 18-layer ResNet3D on UCF101 as a demonstration. Note that we train all networks from scratch without heavy tuning to the hyperparameters or any pretraining on very large datasets due to hardware and time constraints, as well as for fair comparisons.
>
> |     Methods    |  Parameters | Accuracy  |
> |:-------------:|:------:|:------:|
> |  ResNet3D-18 | 33.16M |41.75 |
> |ResNet3D-18 *ACDC-net m=18* | 4.93M | 41.67 |
> |ResNet3D-18 *ACDC-stage m=18* |6.48M | 42.95 |
>
>
> [a] Learning deep features for discriminative localization, CVPR 2016.
>
> [b] SVCCA: Singular vector canonical correlation analysis for deep learning dynamics and interpretability, NeurIPS 2017.

---

### Official Review · Reviewer_FSGC · 2021-11-02

**Correctness:** 4
**Technical Novelty And Significance:** 2
**Empirical Novelty And Significance:** 3
**Recommendation:** 6
**Confidence:** 3

**Main Review:**

[Strengths]

* The idea to use a low-rank decomposition of convolutional kernels and to consequently use the coefficients as sharable weights throughout a network is really interesting. This is especially true for the filter groups networks as the authors show a significant reduction in parameters and improvement in model performance (Table 1). Performance of the method when compared against a good number of _light_ CNN architectures on CIFAR-10/1000 + TinyImageNet is good and further demonstrates the performance.

[Weaknesses]

* The main motivation for the joint subspace (Section 2.2) is quite confusing in the opinion of this reviewer. It is difficult to understand why exactly one can exploit "intrinsic, low-rank structure...across layers" from the explanation and the pilot MNIST experiment. This is an important section that needs to be reworked.

* The idea to share filter basis' across layers is not entirely new some and _shares_ similarites to https://arxiv.org/pdf/2006.05066.pdf. (Deeply Shared Filter Bases for Parameter-Efficient Convolutional Neural Networks). In that paper, they also learn a filter basis that can be shared across layers despite a different methodology.

[Other]

* In the filter groups model, how exactly does ACDC work? Is there a limitation that you need to have equal amounts of filter groups at every layer as the dimension of the shared coefficient is contingent on the groups _g_?

* The decomposition method that is based on Qiu et al. 2018 is presented in Section 2.1. How much does this differ? Why not just use Qiu et al?

* I would double check the notation for all subscripts as some $\times$ symbols seem to be missing

**Summary Of The Paper:**

This paper proposes a novel method to decrease redundancy of parameters in a CNN through a joint subspace view. The authors use a low-rank decomposition of the convolutional kernel (A * D) into shared coefficients (A) and atoms (D). The authors propose a variety of proposal methods for their method termed Atom Coefficient Decomposed Convolution. Notably by sharing A across all layers, sharing A across groups of layers and by utilising the idea of filter groups. The authors demonstrate that ACDC-x outperforms baselines at moderate to significant reductions in model size. Further, they show that their method can alleviate issues of model degradation as deeper networks are used.

**Summary Of The Review:**

The idea to learn a joint subspace view to share weights across layers in a CNN is interesting especially with filter groups. They show good results on a variety of datasets against a solid number of baselines. Despite this, it is to my feeling that the idea is not entirely novel as i) low rank decomposition of convolution kernels has already been performed and ii) the idea to share filter basis/weights across layers has also been proposed. Further, the motivation/pilot experiment for the insight to sharing shared coefficients over layers needs to be worked on. I therefore propose 6 for now.

---

> ### Author Response · Authors · 2021-11-16
> **Thank you for your constructive review!**
>
> Thank you for your constructive comments. The difference compared to DCFNet [Qiu et al. 2018] is presented in the response to all reviewers above. We hope the response below could alleviate your further concerns.
>
> **1. Motivating experiment**
>
> While previous research has shown that there is some channel correspondence across CNN layers, it is difficult to see any low-rankness among filters across layers because the spatial pattern can vary significantly from shallow (low-level clues) to deep (semantic clues) layers. Performing the decomposition as shown in Eq. (1) allows us to decouple spatial patterns (low-dimensional filter atoms) and channel mixing (high-dimensional atom coefficients).
> And, as shown in Section 2.2, performing linear alignments with SVCCA after training a network with atom-coefficient decomposition reveals a clear low-rank structure in coefficients across different layers.
> Specifically, in our experiment of a 32-channel network, after alignments, while the coefficients of a single layer appear to be nearly full-rank (31.98), stacking the coefficients of two layers only slightly increases the rank of the joint coefficient matrix (38.56).
> This low-rank structure suggests that, up to some linear transformations that can be absorbed in the per-layer atoms, the coefficients across layers, which contain the majority of the parameters, can be well approximated by a shared set of coefficients.
> This motivating experiment supports the fact that, rather than simply enforcing shared components in a CNN, our method is actually exploiting the cross-layer low-rank structures that exist inherently in CNNs but were previously hidden using standard training methods.
>
>
> **2. Comparisons against [a]**
>
> Convolutional filters in [a] are decomposed over a set of 3D basis, and the sharing scene is motivated merely by the conjecture of potentially sharable components among filters.
> On the other hand, our method is motivated by observed cross-layer low-rankness in coefficients that are trained independently among layers.
> Furthermore, our method instead decomposes convolutional filters into 2D filter atoms, resulting in a significantly more imbalanced parameter partition (very high dimensional sharable coefficients and very low dimensional layer-specific atoms). Subsequently, our method allows a much wider range of trade-offs between expressiveness and parameter size.
> **On ImageNet, our method can achieve roughly the same performance with [a] while using only about half of the parameters, and can further reduce the parameter to 14% of [a] with a reasonable performance drop.**
> In shorts, compared to [a], our method enjoys the advantages of being more flexible and better motivated by the intrinsic cross-layer redundancy.
>
> **3. How does ACDC-g work**
>
> One of our main goals is to develop a *plug-and-play* replacement for convolution layers that allows for minimal changes to network structures such as the number of channels and layers.
> As a result, ACDC-g does permit different numbers of channels and therefore different amounts of filter groups at every layer.
> Modern CNNs typically have increasing channel numbers, which ACDC-g handles by allowing the output of each shallower layer group to be fed into multiple groups in the following layer, where the larger amounts of channels are partitioned into more groups, and all filter groups across the entire network share the same size to allow efficient coefficient sharing.
> This has been clarified in the revision Section 3.4.
>
> **4. Difference with [Qiu et al. 2018] [b]**
>
> As summarized in the response to all reviewers, DCFNet [b] primarily focuses on the low-rankness of the filters in each layer individually, while our method is motivated by the observed cross-layer channel correspondence and the joint low-rankness among coefficients across layers. While a similar decomposition scheme is used, our method has distinct advantages in terms of sharing flexibility and quantitative performance.
>
>
> [a] Deeply Shared Filter Bases for Parameter-Efficient Convolutional Neural Networks, ArXiv 2020.
>
> [b] DCFNet: Deep neural network with decomposed convolutional filters, ICML 2018.

---

> > ### Comment · Reviewer_FSGC · 2021-11-28
> > **Thanks a lot for your rebuttal!**
> >
> > Thanks a lot to the authors for their response and the updated manuscript. I have read their response, all reviews and the manuscript. With better clarification on the difference with [a], better description on the motivation and extra results, I vote for acceptance.
> >
> > I hope that the authors will release PyTorch code upon publication unless I have missed statements on this

---

> > > ### Author Response · Authors · 2021-11-29
> > > **Thanks for your support!**
> > >
> > > Dear reviewer FSGC,
> > >
> > > Thanks for your support for our paper. We will release the PyTorch code upon acceptance.
> > >
> > > Best regards,
> > >
> > > Authors

---

> ### Comment · Area_Chair_VSUA · 2021-11-26
> **Please respond to the author rebuttal**
>
> Dear Reviewer FSGC,
>
> The authors have posted their rebuttal. I wonder whether the rebuttal addressed your concerns? Please respond to the authors. Thanks!
>
> AC

---

### Official Review · Reviewer_v8NG · 2021-11-07

**Correctness:** 3
**Technical Novelty And Significance:** 2
**Empirical Novelty And Significance:** 2
**Recommendation:** 6
**Confidence:** 5

**Main Review:**

The factorization approach for convolutional layers is exactly that proposed in DCFNet [Qui et al., 2018].  The difference is in terms of choosing to share channel mixing coefficients between multiple layers.  However, [Qui et al., 2018] focus on another means of parameter reduction -- namely, choosing a fixed basis for the filter atoms, and report accuracy-parameter efficiency improvements.

As the paper makes use of the same factorization as DCFNet [Qui et al., 2018], I would expect an experimental comparison between DCFNet and the proposed approached.  The paper also cites [Li et al., 2019] as another previous work that uses a filter basis factorization.  Moreover, Section 4.2 of [Li et al., 2019] proposes sharing the same basis across multiple convolutional layers.  Given this similarity in terms of ideas, it also seems necessary to experimentally compare to [Li et al., 2019] and provide further discussion of how the proposed approach and/or experimental results differ from these two prior works.

Furthermore, though the paper does compare results to LegoNet [Yang et al, 2019] in Table 2, that comparison is uninformative as it applies the filter parameterization scheme of LegoNet to VGG, while applying its proposed scheme to Wide ResNet.  VGG vs Wide ResNet is a significant change in the base network architecture, which is a confound for the question of which filter parameterization is better; each method could be applied to either base architecture and the comparison should be made by choosing the same base architecture.


**Summary Of The Paper:**

This paper proposes a scheme for partially sharing CNN parameter across layers.  Specifically, convolutional layer parameters are factored into the product of layer-specific filter atoms (which define spatial filter structure) and a set of shared coefficient that mix features across channels.  Training CNNs reparameterized in this manner yields improvements along the accuracy-parameter tradeoff curve, compared to baseline models on image classification tasks.

**Summary Of The Review:**

This paper proposes a cross-layer parameter sharing scheme for CNNs that demonstrates some benefits on image classification tasks.  However, the proposed method is similar to filter factorization approaches in several previous published works, for which experimental comparison is missing or incomplete.

Update (after rebuttal):
The author response addresses some of the experimental concerns from my initial review.

---

> ### Author Response · Authors · 2021-11-16
> **Thank you for your insightful review!**
>
> Thank you for your constructive comments.
>
> The discussion of the difference compared to DCFNet [Qiu et al., 2018] is presented in the response to all reviewers. We hope the response below could alleviate your further concerns.
>
> **1. Comparisons against [Li et al., 2019] [a]**
>
> The method of decomposition and sharing differs significantly between our method and [a].
> Each convolutional filter in [a] is decomposed as a low-rank linear combination of **3D** basis elements, and the 3D basis elements are shared among layers.
> On the other hand, as discussed in Q1 to all reviewers, our method is motivated by the intrinsic **cross-layer channel correspondence**, and the subsequent observation of **low-rank structure among cross-layer atom coefficients** after decomposing each convolutional filter over a set of **2D** atoms.
> In our method, we keep the low-dimensional 2D atoms layer-specific, and share the atom coefficients across the network, thus achieving better parameter saving.
> We have included more comparisons against [a] in Table 2 of the revision.
> In comparison to [a], our method achieves better parameter-performance trade-offs.
>
> **2. Comparisons against LegoNet [b]**
>
> To save space, we mainly present in Table 2 results that are not covered by Table 1.
> Some results of applying various variants of ACDC to VGG are already included in Table 1.
> Combining the data from the two tables, it can be seen that when applying to VGG, LegoNet needs **3.7M** parameters to achieve a **6.77** error rate on CIFAR-10, whereas our method needs only **0.13M** parameters to achieve a **6.68** error rate.
> Our method demonstrates clear advantages. We have modified Table 2 in the revision and added the representative comparisons on VGG (marked in blue).
>
> We summarize the performance comparisons in the following table, for more details please refer to the revision of our paper.
>
> |     Methods    |  Parameters | Accuracy  |
> |:-------------:|:------:|:------:|
> |  DCFNet (K=6, FB) VGG16 [Qiu et al., 2018] | 9.84M | 5.98 |
> |Learrning filter bases VGG16 [a] | 4.93M | 6.23 |
> |LegoNet-VGG16-w(o=4,m=0.25) [b] | 0.9M | 8.65 |
> |LegoNet-VGG16-w(o=2,m=0.5) [b] | 3.7M | 6.23 |
> | VGG16 ACDC-g-layerm8s32 | 0.13M | 6.68 |
> |VGG16 ACDC-netm8 | 2.11M| 5.67 |
>
>
>
> [a] Learning filter basis for convolutional neural network compression, CVPR 2019.
>
> [b] LegoNet:  Efficient convolutional neural networks with lego filters, ICML 2019.

---

> ### Comment · Area_Chair_VSUA · 2021-11-26
> **Please respond to the author rebuttal**
>
> Dear Reviewer v8NG,
>
> The authors have posted their rebuttal. I wonder whether the rebuttal addressed your concerns? Please respond to the authors. Thanks!
>
> AC

---

### Author Response · Authors · 2021-11-16
**Thank all reviewers for the constructive comments!**

We thank all reviewers for the supportive and constructive comments.

We first address the shared concerns here, and then address the comments raised by each reviewer individually.
Based on the comments, we have revised our manuscript. Major modifications and titles of new sections are in blue.

**1. Motivation, novelty, and advantages**

We highlight next our contributions that distinguish the proposed method from other tensor factorization based methods.

While preliminary studies, e.g., [a]，suggest the existence of certain channel correspondence across different CNN layers, exploiting this potential redundancy remains challenging. We suspect that the spatial patterns of convolutional filters, which correspond to different semantics levels, are the main reason why we cannot observe obvious cross-layer filter low-rankness. Thus, we test our hypothesis by decoupling spatial patterns (filter atoms) and channel mixing (atom coefficients).
Now, after certain alignments, we can finally observe low-rank structures among atom coefficients across layers, e.g., as observed in Section 3.2, $\text{rank}(\tilde{\mathbf{A}}_i) : \text{rank}([\tilde{\mathbf{A}}_i, \tilde{\mathbf{A}}_j]) = 31.98 : 38.56$.
That is, **concatenation of atom coefficients from two arbitrary network layers almost maintains the same rank of every single layer, instead of doubling.**
This highly non-trivial observation, which has **never been reported before** to the best of our knowledge, suggests a joint subspace view to CNNs, and subsequently **a novel CNN architecture in which the majority of parameters are shared across layers as a common block of atom coefficients, with only a few hundred parameters remaining specific to each layer at filter atoms.** We have incorporated the above discussion into the revised introduction (marked in blue).


This novel network architecture, i.e., a single large block of atom coefficients, together with layer-specific filter atoms (**typically at most a few hundred parameters per layer**), results in significant parameter savings. For typical $3 \times 3$ convolution, for example, using $m=8$, we have only **72** layers-specific parameters left in each layer.
The advantages of our method can be further reflected by the extraordinary flexibility as presented in Table 1, where we show that ACDC can reduce the parameters of a VGG network to **0.4%** of the original model, while still delivering over **90%** accuracy on CIFAR-10. Even on those relatively small-scale datasets, we believe the results produced by our method are still impressive.


Beyond maintaining performance and reducing model size, we further show additional advantages of this novel network architecture by demonstrating its *easier model adaptation* (Section 4.2) and *better interpretation* (Section 5).

**2. Comparisons against DCFNet [b]**

The decomposition introduced in DCFNet is primarily driven by the low-rankness of the convolutional filter in **each individual layer**, and is used to impose filter regularity via bases truncation, with the number of used basis elements determining the degree of parameter reduction.
As shown in the motivating experiment presented in Section 2.2, our method is primarily driven by the joint subspace view to CNNs motivated by the underlying **cross-layer channel correlations**, as opposed to DCFNet.
ACDC has a clear advantage in terms of performance and parameter reduction by providing a much wider range of trade-offs between accuracy and model size.
ACDC can achieve **99.6%** parameter reduction when applied to VGG and CIFAR-10 experiments, which is remarkably better than DCFNet, which can only reduce the parameter by **44.4%**. We have updated Table 2 of the revision to include the comparisons against [b].

[a] SVCCA: Singular vector canonical correlation analysis for deep learning dynamics and interpretability, NeurIPS 2017.

[b] DCFNet: Deep neural network with decomposed convolutional filters, ICML 2018.

---

### Author Response · Authors · 2021-11-25
**Thank all reviewers for the supportive and constructive feedback!**

Dear reviewers,

Once again, we would like to express our gratitude to all reviewers for their supportive and constructive feedback.

We have addressed the following comments and updated our manuscript accordingly:
- We have clarified the motivation, novelty, and advantages of our method in the response to all reviewers, and updated the Introduction section in the revision.
- We have discussed the difference and advantages of our method compared to [a,b,c,d] in terms of motivations, technique details, and empirical results. The Experiment section of our manuscript was updated accordingly.
- We have conducted further experiments on extending the proposed method to 3D convolutions as suggested by Reviewer **FvfC**.
- To demonstrate how to determine the level of parameter sharing suggested by Reviewer **FvfC**, we have developed a primary method for adapting the model architectures permitted by the significant flexibility of our method, and presented results in Section E of revision.
- We have presented further clarifications regarding details of ACDC-g, interpretability, network speeding-up, and the motivating experiment.
- We have updated the tables, captions, and fixed the typos in the manuscript.
- As suggested by Reviewer **r8DN**, we have reorganized the presentation of the paper.

As the final deadline for the discussion period is approaching, we sincerely hope that reviewers will read our response again to see if there are any further concerns that we can address.

Thank you,

Authors

___

[a] DCFNet: Deep neural network with decomposed convolutional filters, ICML 2018.

[b] Learning filter basis for convolutional neural network compression, CVPR 2019.

[c] LegoNet: Efficient convolutional neural networks with lego filters, ICML 2019.

[d] Deeply Shared Filter Bases for Parameter-Efficient Convolutional Neural Networks, ArXiv 2020.

---

### Author Response · Authors · 2021-11-27
**Message for reviewers**

Dear reviewers,

Given only two days left for the discussion period, please kindly let us know if any additional concerns we might be able to address. So many thanks!

Best regards,

Authors

---

### Comment · Area_Chair_VSUA · 2021-11-28
**Please post your post-rebuttal opinions!**

Dear Reviewers,
The authors have updated their manuscript and responded to your comments. Please check whether your concerns have been addressed and then post your further opinions *if you haven't*. This is the professional way to show respect to the authors' efforts. The deadline Nov. 29 is coming very soon. Thanks!

AC

---

### Author Response · Authors · 2021-11-29
**Thanks to all reviewers for the comments and discussion**

Dear reviewers,


We sincerely appreciate all reviewers for the unanimous support and their insightful feedback that helps greatly improve our manuscript.

The source code of our paper will be released upon acceptance.


Best regards,

Authors

---

### Decision · Program_Chairs · 2022-01-20

**Decision:**

Reject

**Comment:**

The paper eventually got 5 "marginally above the threshold" after rebuttal. Such scores testify to that the paper is a borderline one. By reading the post-rebuttal comments, it is evident that most of the reviewers still deemed that the novelty is incremental. One of the reviewer (vUb9) raised the score simply to "encourage the authors to think more important problems", rather than acknowledging the merits of the paper. The AC also read through the paper and had the following opinions:
1. The paper is actually about DNN compression, based on the "new finding" that the weights across layers are low-rank. However, the authors would not write the paper in the way of DNN compression, but put more emphasis on the "new finding", which has no theoretical support at all (only some heuristic reasoning). The AC would deem that the "new finding" is only an assumption.
2. Actually the "new finding" is not new at all. For example,

[*] Zhong et al., ADA-Tucker: Compressing Deep Neural Networks via Adaptive Dimension Adjustment Tucker Decomposition, Neural Networks, 2019,

used a shared core tensor (which could be regarded as the common dictionary) across all layers for higher compression rates. More recent references that use tensors and consider shared information across layers for compression can be easily found as well.

So the AC thanked the authors for preparing the rebuttals carefully, but regretfully the paper is not good enough for ICLR.